# Environmental Benefits of Stock Evolution of Coal-Fired Power Generators in China

**Fangyi Li [1,2]**, **Zhaoyang Ye [1,2]**, **Xilin Xiao [1,2,\*]** and **Dawei Ma [3]**

[1] School of Management, Hefei University of Technology, Hefei 230009, China; fyli@hfut.edu.cn (F.L.); 2017110808@mail.hfut.edu.cn (Z.Y.)

[2] Research Center of Industrial Transfer and Innovation Development, Hefei University of Technology, Hefei 230009, China

[3] Power Technology Centre, State Grid Anhui Electric Power Corporation Research Institute, Hefei 230601, China; dwma@mail.ustc.edu.cn

\* Correspondence: 2018110773@mail.hfut.edu.cn; Tel.: +86-551-6291-9155

**Abstract:** The evolution of in-use coal-fired power generators (CPGs) in China has been impelled by a series of policies called *Developing Large Units and Suppressing Small Ones* in recent decades. However, it remains highly uncertain about the contribution of the evolution on air pollution reductions at different stages. Models used to assess the effects of CPGs' evolution often do not account for the different boundary conditions related to units composition and age structure of the existing CPGs, and lifetime expectancy, which hinders effective strategy development and realistic target setting. This study employs a dynamic Type-Cohort-Time (TCT) stock-driven model and Logarithmic Mean Divisia Index (LMDI) technique, to investigate the structural evolution of China's CPGs as well as its environmental effects from 1980 to 2050. We consider generator-specific characteristics, lifetime-related issues and alternative techniques in the historical and scenario analysis. The main results are as follows: the environmental benefits of structural evolution were limited, compared with the changes in emission coefficient due to technical renovation. However, scenario analysis indicates that structural adjustment by elimination of outdated CPGs and construction of new ones in future will undertake emission reduction commitments, since the potentials of other approaches decrease. Uncertainty analysis further demonstrates that promoting elimination of small CPGs and substituting them with renewable energy will bring more emission reductions. The key findings can support policy-making on elimination, construction, and emissions abatement of CPGs.

**Keywords:** CPG; environmental benefits; type-cohort-time; LMDI; scenario analysis

## 1. Introduction

The growth in electricity demand and its environmental consequences, especially in emerging economics, has attracted widespread scientific attention [1,2]. A growing amount of coal- or gas-based power plants around the world will commit large amounts of $CO_2$ emissions [3]. Since China started to focus on economic development in the 1980s, the power sector has made great progress. The installed capacity of power generation increased to 1900 GW (109 watts) in 2018, in which 1144 GW is thermal power generation [4]. Although the power system sustained social and economic development, its impacts on climate change, atmospheric environment [5], and accompanying threats to public health [6,7] cannot be ignored [8–10]. In order to abate the environmental damage of the power sector, China not only implemented great investment in decontamination of coal-fired power generators (or CPGs), including dedusting, desulfuration, and denitration, but also strived to promote elimination of outdated CPGs and construction of new ones. The former was a common understanding and

made a great achievement. The later, summed up as *Developing Large Units and Suppressing Small Ones*, succeed in accelerating structure evolution of CPGs and application of new technologies, but resulted in overheated investment and over capacity. In the latest measures, more than 150 GW of coal power projects will be cancelled or postponed to 2020. The installed coal power capacity will be constrained less than 1100 GW and about 20 GW of outdated CPGs will be eliminated by 2020 [11]. It is of great significance to investigate the structure evolution of CPG stock as well as accompanying environmental effects in the past and future, thus relevant policies can be reassessed. Besides, the increasing installed capacity of renewable energy squeezed the development space of thermal power and generated environmental benefits [12,13], which should be involved in such studies. In the paper, we analyze the structure evolution of China's CPG stock from 1980 to 2018 with Type-Cohort-Time (TCT) approach, and estimate the environmental benefits of structure evolution in history and future with the Logarithmic Mean Divisia Index (LMDI) method.

The TCT model is able to handle the complexity and heterogeneity of stock evolution by breaking down the inventory into its basic dimensions, which are characterized by the different sizes, driving forces, and techniques [14]. Environmental impacts of stock evolution are of concern for strategy and policy-making as important references. There are three perspectives when estimating environmental impacts of stock change. First, estimating direct effects for a continuous period like 10 years or 50 years. Then, the changes in environmental effects in the period can be decomposed into several driving forces by LMDI method or other kinds of models. Similar studies in decomposition of change in energy consumption [15] and $CO_2$ emissions [16–18] have been implemented in different countries and regions. Second, estimating accumulative effects of stock for a long period or in a special process, which is directly related to environmental or climate outcomes. For example, the cumulative future emissions of currently operating and planned infrastructures are much more relevant to climate outcomes than individual annual emissions of assets [19]. This approach is appropriate to compare schemes with different infrastructures or equipment in a given period. Third, estimating life-cycle effects for the whole life of the objects, which have attracted increasing attention in recent years. For example, the life-cycle emissions of thermal power plants were estimated in different countries and different scenarios [20–24]. Ning [25] found that the carbon emissions in the life cycle are higher than the direct emissions of coal-fired power generation by 10%–13% in China. Odeh [26] found that most of the carbon emissions in the life cycle come from the operational stage (89%) in the UK. Different power technologies were assessed from the perspective of life cycle [27,28], such as wind power [29–31], photovoltaic system [32], and hydropower [33]. New technologies for low-carbon development, like conversion of carbon dioxide into clean fuels, can reduce the life-cycle impacts on environment [34]. In addition, life-cycle assessment can also be applied in scenario analysis for macro level analysis [12,35–37].

In recent years, the impacts of coal-fired power generation on $CO_2$ emissions and global warming has attracted widespread attention from many scholars [2,3], while its contribution to air quality improvement is also an important issue [10,38,39]. Previous studies were designed to reveal the driving forces of impacts' changes on a global scale [40], national scale [38,39], regional scale [41] and microgrid scale [42], with different dynamic mechanisms. Most dynamic models applied for future evaluation and prediction are scenario analysis based on plans and policies [37], computable general equilibrium [43], multi-objective optimization [44], and dynamic linkage [45]. However, it remains highly uncertain about the contribution of the evolution on air pollution reductions at different stages. Models used to assess the effects of CPGs' evolution often do not account for the different boundary conditions related to units composition and age structure of the existing CPGs, and lifetime expectancy, which hinders effective strategy development and realistic target setting. The evolution of CPG stock and its causes, as well as the resulting environmental impact, cannot be clearly demonstrated.

To investigate and demonstrate the structural evolution of China's CPGs in recent decades and in future, we establish a dynamic TCT stock-driven model of CPGs, with consideration of generator-specific characteristics, lifetime, and alternative technique. In the historical analysis,

Logarithmic Mean Divisia Index (LMDI) technique is applied to identify the environmental effects of various driving forces, including structural evolution. In the scenario analysis and uncertainty analysis, impacts of a generator's lifetime and substitution power on pollutant emissions are detected. The main contributions of this study are as follows: (1) to reveal the characteristics and trends of stock evolution of CPGs in China from 1980 to 2050; (2) to demonstrate the environmental effects of CPGs evolution and identify the driving forces in different periods; and (3) to identify the environmental effects of lifetime change of small CPGs, which corresponds to the policies called *Developing Large Units and Suppressing Small Ones*.

The rest of this paper is as follows: the second section mainly describes the method and data sources of this study, the third section presents the results and discussion, and the fourth section shows the conclusion and suggestion.

## 2. Method and Data Sources

### 2.1. TCT Model

The TCT model is often used in material stock and flow analysis, which is employed to reveal the stock evolution of CPGs in this study. It takes the scrap rate of different units into account and establishes a dynamic mechanism, to demonstrate the evolution of China's CPGs in the past and future. First, a system was established in this study, as shown in Figure 1. The CPG stock was differentiated using the types and cohort groups. Second, a dynamic stock-driven model was developed for tracking and predicting flows of CPGs through all type-cohort fractions of the stock. Third, stocks and flows in future were used to calculate various emissions by CPGs from 2019 to 2050.

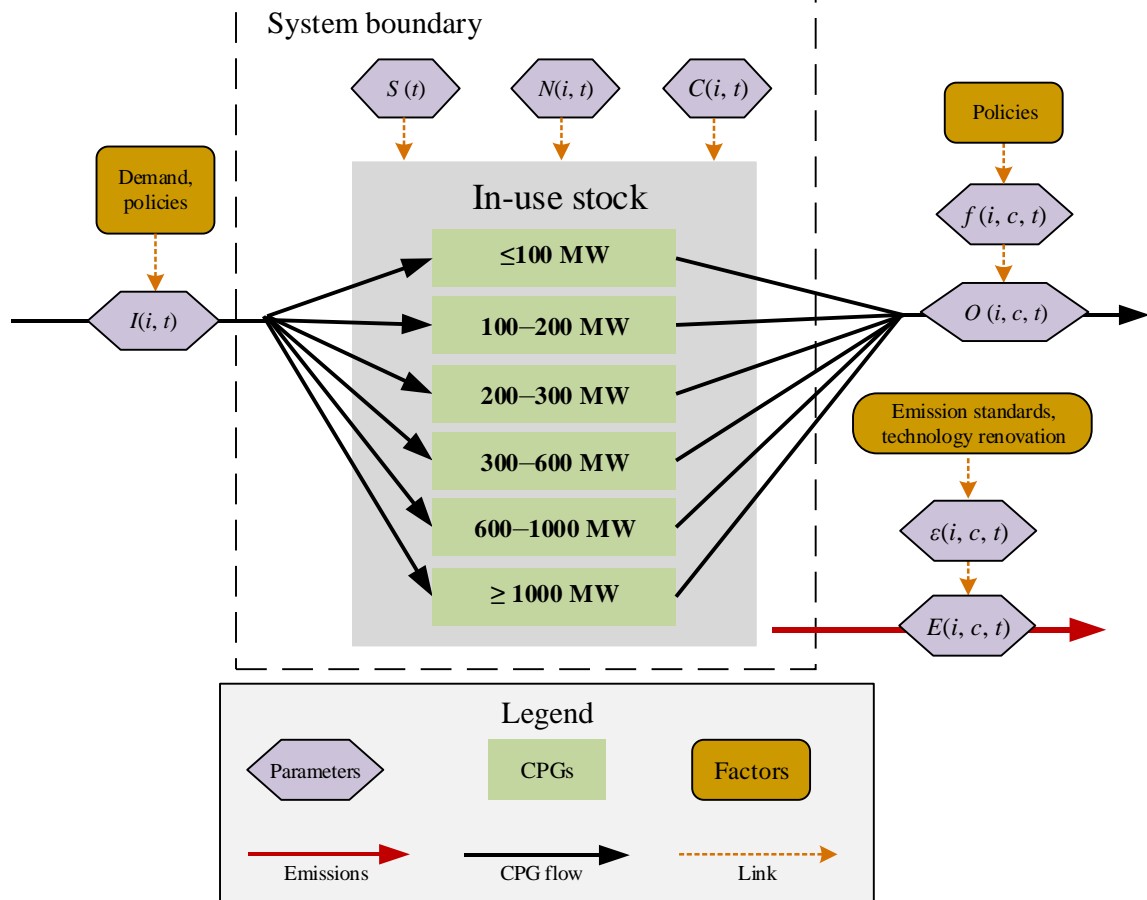

**Figure 1.** System boundary and model definition. CPGs: coal-fired power generators.

According to the literature [14], three aspects are emphasized in TCT analysis: (i) types of CPGs, (ii) cohorts, and (iii) time. Types and groups correspond to the structure (typology) of CPGs. Time refers to the modeling time range—one year or several years—and whether this corresponds to historical, current, or future situations. Model types can be divided into three categories: (i) calculations, (ii) quasi-static, and (iii) dynamics. The computational model primarily quantifies the size and composition of the CPG stock, as well as the associated matter or energy flow. Quasi-static models usually study the stock of CPGs for one year, while dynamic models are analyzed for many years. At present, the stock calculation method adopted by the mainstream is the bottom-up model.

In the system, the technical level is defined by installed capacity of one CPG ($C_{i,t}$), where $i$ represents the level and $t$ represents the year. Therefore, the total size of CPG stock ($S_t$) can be calculated by the sum of all levels as shown in Equation (1). $N_{i,t}$ is the number of CPGs of level $i$.

$$S_t = \sum_{i=1}^{m} N_{i,t} \times C_{i,t} \tag{1}$$

The annual change of CPG stock ($\Delta S_t$) is the difference between the current year's stock ($t$) and the previous year's stock ($t-1$).

$$\Delta S_t = S_t - S_{t-1} \tag{2}$$

The total construction of a year is the sum of the net change in stock ($\Delta S_t$) and the substitution of the CPGs eliminated ($O_t$). The construction of a specific type of CPG ($I_{i,t}$) is defined by the type split-share ($TS_{i,t}$) as in the following equation.

$$I_{i,t} = (\Delta S_t + O_t) \times TS_{i,t} \tag{3}$$

If $\Delta S_t + O_t > 0$ in a given year, the construction can be calculated. If $\Delta S_t < 0$ and $\Delta S_t + O_t < 0$, $I_t < 0$ makes no sense and should be avoided. It means no new installed capacity is needed, so we make $I_t = 0$. Then, the adjusted installed capacity of eliminated CPGs ($O_t'$) is calculated as Equation (4).

$$O_t' = |\Delta S_t| = |S_t - S_{t-1}| \tag{4}$$

The demolition of a type-cohort combination in a specific year is a function of the size of each type group ($I_{i,c}$) and its probability of reaching the end-of-service-life ($f_{i,c,t}$) given its age. $f_{i,c,t}$ is derived from the probability density function of a normally distributed lifetime, as shown in Equation (6). $c$ represents the year of the CPG into use:

$$O_{i,c,t} = I_{i,c} \times f_{i,c,t} \tag{5}$$

$$f_{i,c,t} = \frac{1}{\sigma\sqrt{2\pi}} e^{-\frac{[(t-c)-\tau]^2}{2\sigma^2}} \tag{6}$$

where $\sigma$ presents the standard deviation of life, $\tau$ presents the average life of the CPGs.

The in-use stock ($S_{i,c,t}$) can be calculated by sum of initial stock $S_{i,1,1}$ and cumulative changes, as shown in Equation (7).

$$S_{i,c,t} = S_{i,1} + \sum_{c=1}^{t} I_{i,c,t} - O_{i,c,t} \tag{7}$$

The total emission in a given year is the sum of the individual emission of the different parts (type-cohort combinations or segments) of the in-use stock ($S_{i,c,t}$). As shown in Equation (8), $\varepsilon_{i,c,t}$ represents emission coefficient, and $T_{i,c,t}$ represents annual running hours.

$$E_t = \sum_{i}^{m} \sum_{c=1}^{t} \varepsilon_{i,c,t} \cdot S_{i,c,t} \cdot T_{i,c,t} \tag{8}$$

### 2.2. LMDI Method

The LMDI method is employed to identify contributions of various driving forces. Based on the LMDI model [46,47] and data availability, the changes of pollutant emissions of China's CPGs are decomposed into different factors. Specifically, pollutant emissions can be written as the product of multiplying installed capacity, installed structure, running hours, and emission coefficient, as shown in Equation (9):

$$E_t^q = \sum_{i=1}^{n} S_t \times \frac{S_{i,t}}{S_t} \times T_{i,t} \times \varepsilon_{i,t}^q \tag{9}$$

where, $E_t^q$ is emission of a specific pollutant ($CO_2$, $SO_2$, $NO_X$ or PM), and $q$ is the type. $S_t$ represents the total installed capacity in $t$ year, $S_{i,t}$ represents the installed capacity of CPGs of level $i$, $T_{i,t}$ represents the annual running hours of CPGs of level $i$, $\varepsilon_{i,t}$ represents emission per power generation of CPGs of level $i$. $R_{i,t} = \frac{S_{i,t}}{S_t}$ is expressed as the installed structure.

According to the LMDI method, from the 0-th to the $t$-th, the change in one type of emission can be expressed as:

$$\Delta E = \Delta E_S + \Delta E_R + \Delta E_T + \Delta E_\varepsilon \tag{10}$$

The right part of the equals sign represents the contribution of five factors from the 0-th to the $t$-th period to the emission change, which are the change in total installed capacity $\Delta E_S$, change in CPG structure $\Delta E_R$, change in annual running hours $\Delta E_T$, change in emission coefficient $\Delta E_\varepsilon$. The expressions for each factor are shown in Equations (11)–(15).

$$\Delta E_S = \sum_{i=1}^{n} w \ln \left( \frac{S_{i,t}}{S_{i,0}} \right) \tag{11}$$

$$\Delta E_R = \sum_{i=1}^{n} w \ln \left( \frac{R_{i,t}}{R_{i,0}} \right) \tag{12}$$

$$\Delta E_T = \sum_{i=1}^{n} w \ln \left( \frac{T_{i,t}}{T_{i,0}} \right) \tag{13}$$

$$\Delta E_\varepsilon = \sum_{i=1}^{n} w \ln \left( \frac{\varepsilon_{i,t}}{\varepsilon_{i,0}} \right) \tag{14}$$

where:

$$w = \begin{cases} \frac{E_{i,t} - E_{i,0}}{\ln \left( E_{i,t} / E_{i,0} \right)}, & E_{i,t} \neq E_{i,0} \\ E_{i,t}, & E_{i,t} = E_{i,0} \\ 0, & E_{i,t} = E_{i,0} = 0 \end{cases} \tag{15}$$

### 2.3. Data Sources

The installed capacity of CPGs of each level in the study period comes from *Compilation of Statistics of the Power Industry in China* [48]. Each year's extension assembly data and various renewable energy generation data are from the China Electricity Council [49]. The pollutant emission coefficient of the CPGs comes from the State Grid Corporation (See Figure A1 in Appendix A). The annual running hours of various energy generation systems are derived from the *China Electric Power Yearbook* [50].

## 3. Results and Discussion

### 3.1. Stock Dynamics

In 2018, the total installed capacity of thermal power generation in China reached 1144 GW [4], produced $5.07 \times 10^{12}$ kWh, including power generation from firing coal, oil, gas, garbage, and biomass [51]. Installed capacity of CPGs reached 1010 GW [49].

For the type and cohort analysis of CPGs, we use level of installed capacity to distinguish individual generators. The evolution of CPG stock from 1980 to 2018 is shown in Figure 2. Generators under 300 MW accounted for the majority of CPG stock in 1980, the proportion of which decreased in the study period. The installed capacity of CPGs of 100 MW and below kept increasing, but the proportion decreased. These CPGs were constructed in undeveloped and remote areas because of their low cost for construction and convenience in technology acquisition. The capacity of 100–200 MW and 200–300 MW CPGs reached an inflection point around 2005, then started to decrease. This was because the widespread use of medium and large CPGs has caused 100–300 MW CPGs to lose their own advantages. Although the existence of 600–1000 MW CPGs was later than 300–600 MW CPGs, their installed capacities increased together after 2005. These two types of CPGs played a dominant role in coal-fired power for a long time. The 1000 MW and above CPGs started to develop in 2006, with a high speed. Compared with small-scale effect, large and extra-large CPGs can save energy and reduce emissions. Extra-large CPGs represent the future standardization of coal-fired power technique, which are encouraged by current policies. So the proportion of CPGs of 1000 MW and above will continue to expand in future. Finally in 2018, 300–600 MW CPGs accounted for 35.7%, 600–1000 MW CPGs accounted for 34.0%, and extra-large CPGs (≥1000 MW) accounted for 8.9%, respectively. The stock evolution before 2010 followed the laws of technological evolution and equipment depreciation, as policies about elimination of backward generators were not implemented.

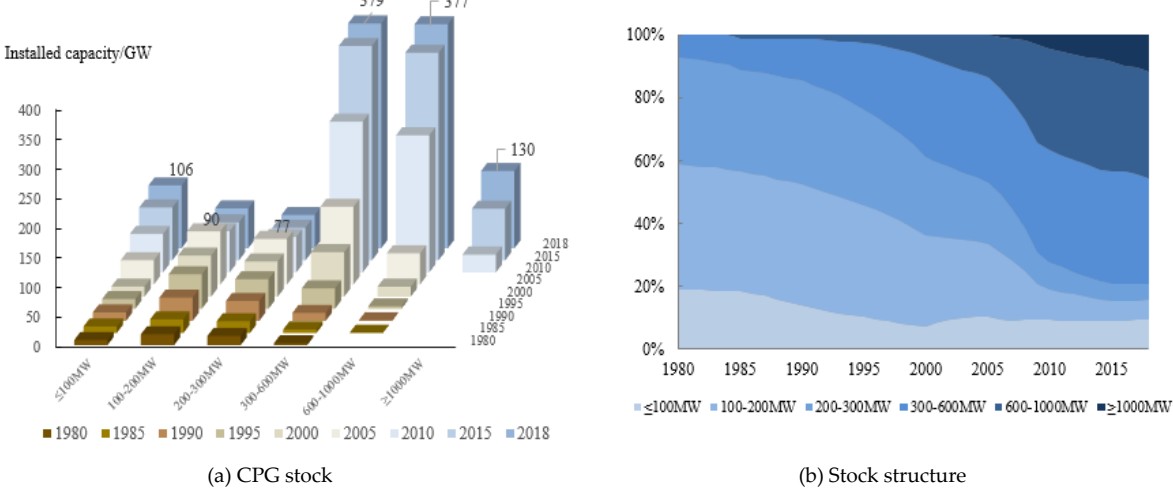

(a) CPG stock　　　　　　　　　　　　　　　　　　(b) Stock structure

**Figure 2.** Evolution of CPG stock from 1980 to 2018.

Since the state council issued the notice on further strengthening the elimination of backward generators in April 2010, the elimination process has been promoted by governments. Through these policies, the power sector is expected to improve efficiency and reduce emissions of air pollutants. According to an official report [52], the power industry actually eliminated 16.9 GW of backward CPGs in 2010, 7.84 GW in 2011, 5.51 GW in 2012, 5.44 GW in 2013, 4.86 GW in 2014, and 5.27 GW in 2015. Some of the eliminated CPGs passed their design lifetimes, some did not.

Stock structure of CPGs varies in different provinces and cities is illustrated in Figure A2. In Hainan and Hebei, 300–600 MW CPGs installed capacity is the largest. In provinces like Guizhou, Yunnan, Anhui, and Jiangxi, where the economy is not developed and there are certain fossil energy resources,

600–1000 MW has the largest installed capacity in the total. However, some regions still maintain a considerable proportion of small CPGs of 100 MW and below (such as Heilongjiang, Shandong, and Tibet. The 100 MW–200 MW and 200 MW–300 MW CPGs account for a very small proportion in most provinces and cities in the country, but it is different in Beijing and Tibet.

The CPGs are designed to service for 30–35 years in China. The service time of existing CPGs till 2018 is shown in Figure 3. Most CPGs were constructed in last 15 years, especially the CPGs above 300 MW. Some CPGs below 300 MW have serviced for 20–30 years, which will be the focus of elimination policies in upcoming years.

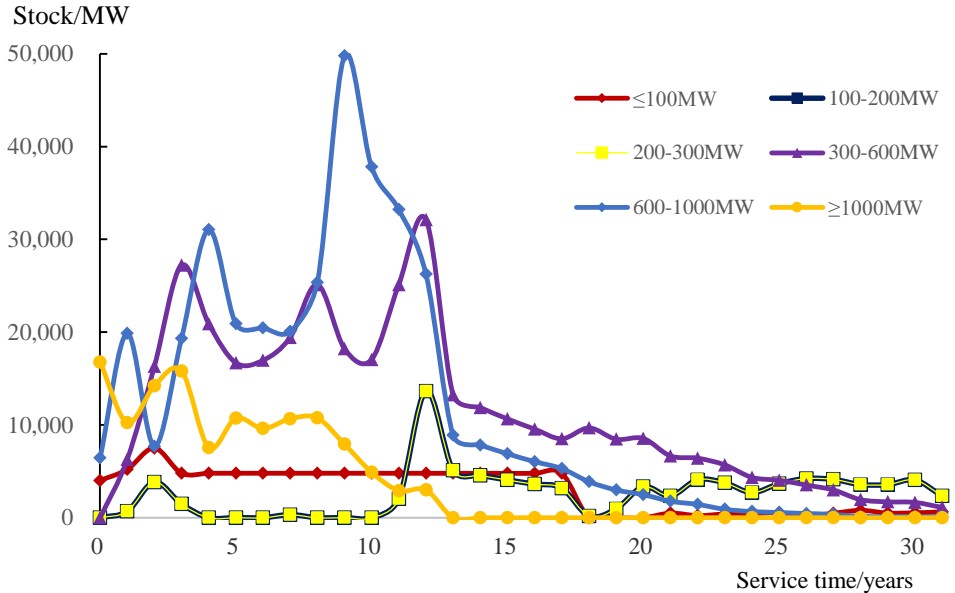

**Figure 3.** Service time of existing CPGs till 2018.

### 3.2. Decomposition of Emission Changes

The trend of pollutant emissions from China's CPGs from 1980 to 2018 is shown in Figure 4. With the increase in the installed capacity of CPGs and the coal consumption, the corresponding $CO_2$ emissions continue to rise. $SO_2$ emissions showed an increasing trend before 2006, with a peak of 13.2 million tons (Mt) in 2006. Subsequently, *Thermal Power Plant Air Pollutant Emission Standards* (GB13223-2003) gradually showed the effect of $SO_2$ control on existing units. With the implementation of the most stringent standards in GB13223-2011 and the ultra-low emission requirements in 2014, the existing CPGs at the end of 2015 completed the upgrading of desulfurization facilities, and improving the operational management. $SO_2$ emissions rapidly decreased from 6.20 Mt in 2014 to 1.6 Mt in 2018, accounting for only 12% of the peak in 2006. $NO_X$ control requirements were relatively loose before 2011. It increased significantly with the increase of coal-fired power generation, and reached the peak of 11.07 Mt in 2011. With the implementation of ultra-low emission requirements and increasing capacity of China's flue gas denitration equipment, the $NO_X$ emissions in 2015 decreased by 71% compared with 2014. $NO_X$ emissions were about 1.45 Mt in 2018, accounting for only 13% of the peak in 2011. The PM emissions showed an overall reduction trend. In 1990 and 2006, respectively, the dust removal technology and the GB1323-2003 standard were promulgated and implemented. The existing CPGs basically completed the first environmental protection technology transformation in 2006. The PM emissions of the coal-fired power sector showed an inflection point in 1991. With the implementation of ultra-low emission limits, China's coal-fired power industry PM emissions were about 0.36 Mt in 2018. In 2015, the State Council introduced the *Work plan of comprehensive implementation of ultra-low emission and energy-saving renovation project for coal-fired power plant*, which required that the CPGs in the eastern, central, and western regions should achieve ultra-low emissions targets at the end

of 2017, 2018, and 2020, respectively. That is, the emission concentrations of PM, $SO_2$, and $NO_x$ are lower than 10 mg/m$^3$, 35 mg/m$^3$, 50 mg/m$^3$, respectively.

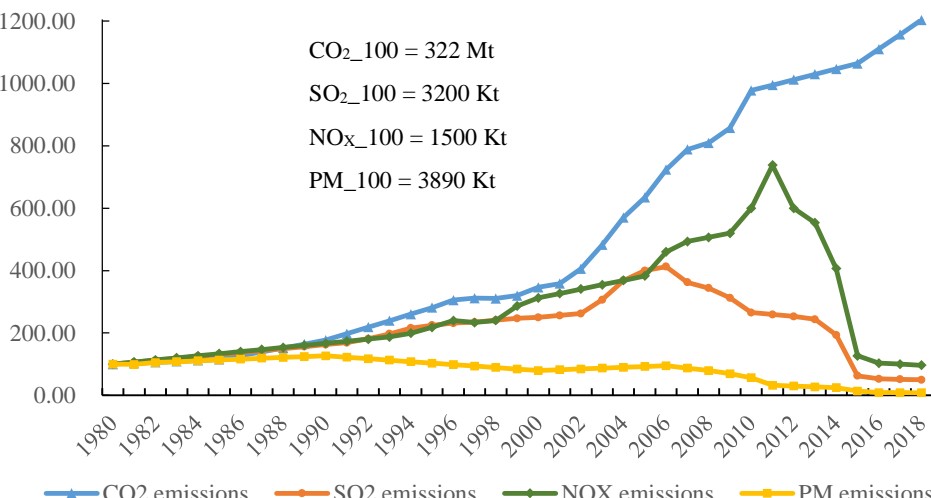

**Figure 4.** Emission changes from China's coal-fired power sector from 1980 to 2018. (Note: Kt stands for thousand tons).

The driving forces of emission changes are identified, as shown in Figure 5. From the overall perspective of various pollutant emissions, the increase in installed capacity is the main reason for emission increase. The emission coefficient is the main reason for emission reduction. This mainly indicates that further improvement of power generation technology will indeed reduce emissions. The emissions reduced sharply following implementation of the corresponding national emission reduction measures, which proved that the national policy played a significant role in reducing pollutant emissions. The change in running time has different effects on emissions of various pollutants, mainly because of the irregular changes in the annual running time, as well as the change in installed structure. The change in emission coefficient (technology improvement) has less impact on $CO_2$ emissions than the impact on other pollutants, as coal consumption reduction and $CO_2$ capture and storage is difficult. For $SO_2$, the change in installed structure and running time have little effect on it, while the emission coefficient is the main factor for its emission reduction. It is mainly because of the implementation of the corresponding GB (National emissions standard) and the development of desulfurization technology. For $NO_X$ and PM, the emission coefficient is also the main reason for its emission reduction, due to the implementation of the new standard and the technology improvement in denitration and dust extraction.

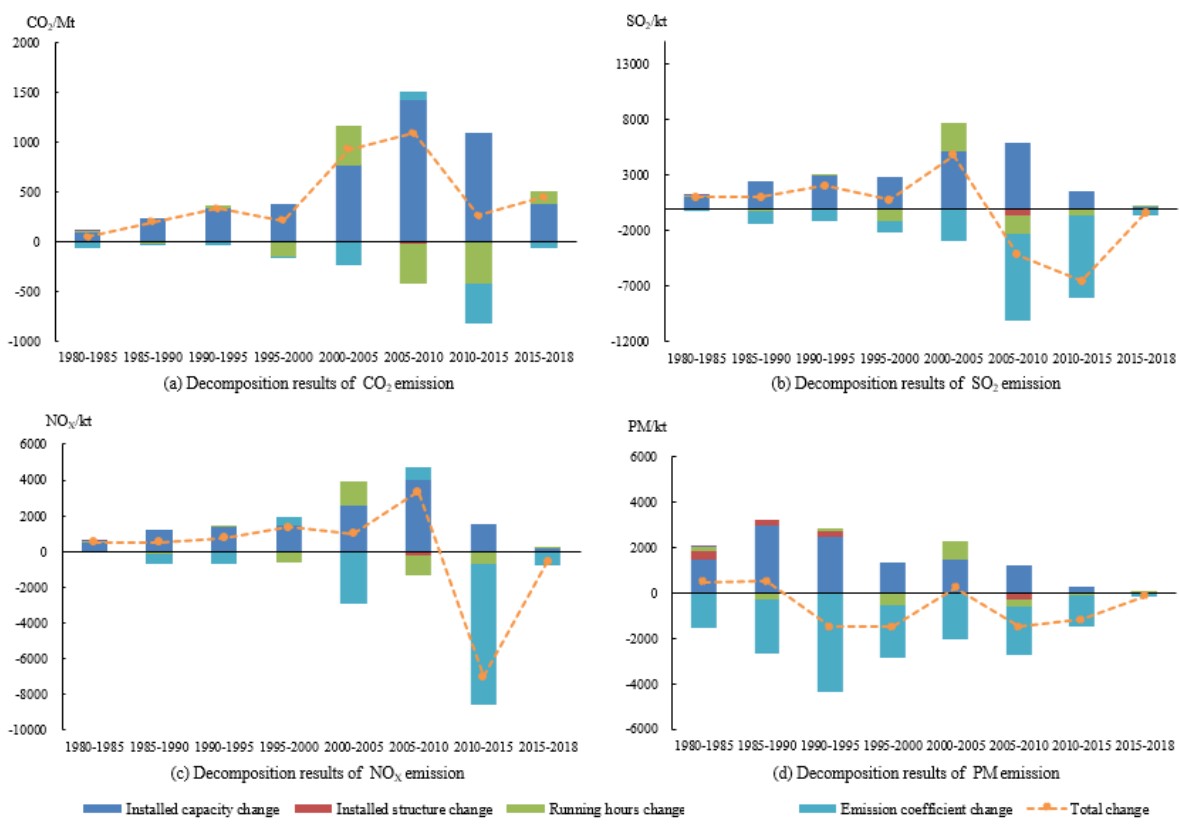

**Figure 5.** Decomposition results of emission change of CPGs.

Corresponding to the analysis results, actual evidence shows that improvement of generation technology is crucial in efficiency improvement [53,54]. As declared by government, China will have upgraded about 420 million kilowatts of CPGs with ultra-low emissions by 2020, the average coal consumption of CPGs nationwide will drop to 310 g of standard coal per kilowatt hour; all generators with a capacity of more than 300 MW have achieved ultra-low emissions [11].

The results also correspond with previous studies. Wang et al. [39] found that treatment technology, power generation technology, and power structure are the main inhibitory drivers, while power demand and population are the promotive ones. Yan et al. [55] also found that technology improvement is the main driver of emission reduction in China's Beijing–Tianjin–Hebei region.

### 3.3. Scenario Analysis

#### 3.3.1. Scenario Design Based on Policies

Scenario analysis is employed to predict the possible impact of future changes in the CPGs structure on carbon and pollutants emissions. Scenarios are built based on existing policies and plans, and TCT dynamic model, with consideration of elimination of backward CPGs and substitution of cleaner power generators (wind, light, hydro power plants, or larger thermal power plants). China's recent CPGs policy and plans are shown in Figure A3. According to references to the National *13th Five-Year Plan for Power Development* [56] and *China Energy Outlook 2030* [57], including the development goals of CPGs in 2020 and 2030 (1040 GW and 1020 GW) and the development goals of related renewable energy, a baseline scenario is established. Other scenarios are built based on the baseline scenario, which differs in total installed capacity, lifetime of specific CPGs, and structure of input generators.

According to the growth rate of electricity consumption in the whole society in the past ten years and the development of energy-saving technologies, the growth rate of power generation in China will be set at 2% before 2030, and power generation will stop increasing after 2030. As lifetime

of infrastructures is a key factor of the outflow and total effects [58], it is selected as a key factor. The lifetime of CPGs is set at 30 years. In addition, inflow of new generators is required to offset the outflow. One approach is developing renewable energy in preference to large-scale CPGs, and the other is the opposite. We set a ratio of installed capacity of large CPGs and renewable energy generation at 50:50 before 2030, and the inflow will be all renewable energy after 2030. Finally, the baseline scenario was established and is shown in Table 1. The key factors have other options for designing other scenarios.

**Table 1.** Scenario design.

| | Growth Rate of Power Generation before 2030 | Service Life of CPGs below 300 MW | Large CPGs: Renewable Energy (Substitution Power Structure before 2030) |
|---|---|---|---|
| Baseline scenario | 2% | 30 years | 50:50 |
| Other options | 1% or 4% | 25 years or 35 years | 25:75 or 75:25 |

### 3.3.2. Stock and Emissions Evolution

The dynamic evolution of CPG stock during 2018–2050 in the baseline scenario is shown in Figure 6. The 300 MW and below generators will be phased out in the next few years, and generators of 300–600 MW will continue to decrease, while those of 600–1000 MW and extra-large (≥1000 MW) will become the absolute main part of CPG stock in the future.

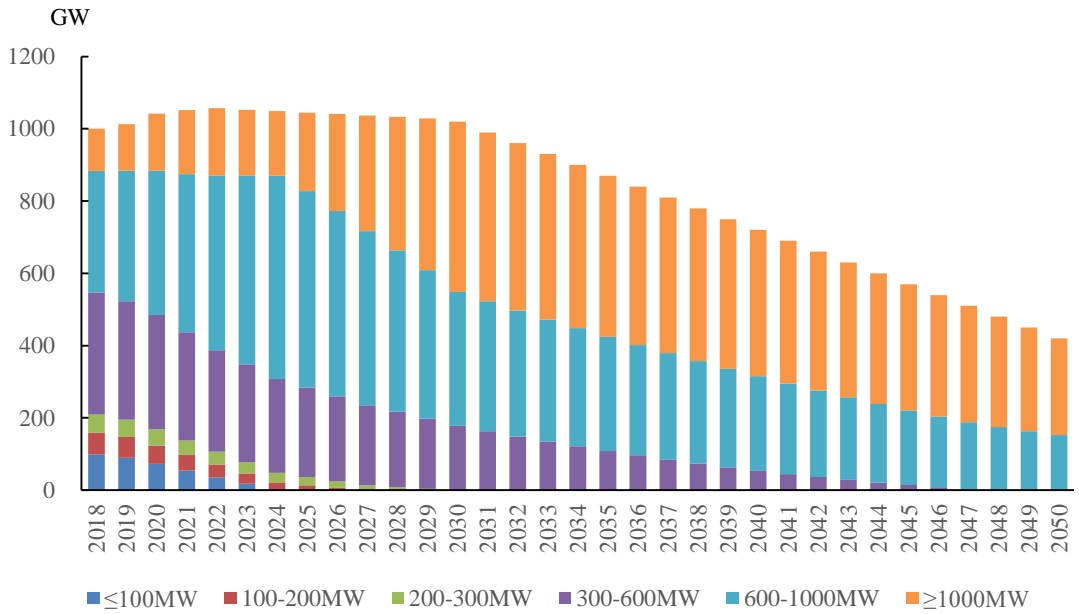

**Figure 6.** Dynamic evolution of CPG stock in baseline scenario.

After the improvement of large-scale emission technology and the implementation of emission reduction policies, it is difficult to make a big breakthrough in the coal consumption and emission coefficient of CPGs. The installed structure will continue to change by increasing 600 MW and above CPGs, and elimination of backward ones, which will become the major driver of emission reductions.

Pollutant emissions from 2018 to 2050 in baseline scenarios are calculated, as shown in Figure 7. The overall trend is that $CO_2$ emissions will increase in the next few years, then will decrease to 1479 Mt in 2050, $SO_2$ emissions dropped from 2718 Kt in 2018 to 799 Kt in 2050, and $NO_X$ from 3832 Kt to 1186 Kt, PM from 2540 Kt to 177 Kt.

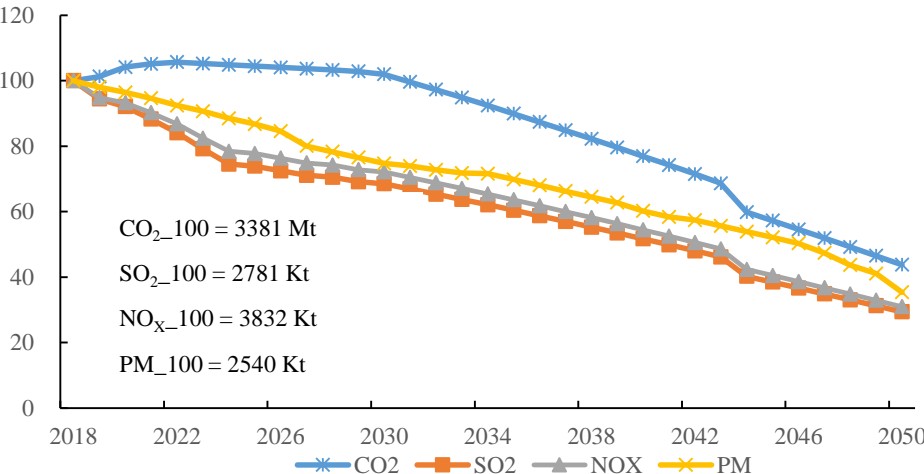

**Figure 7.** Annual emissions of CPGs in baseline scenario.

### 3.3.3. Uncertainty Analysis of Scenarios

The data and results of scenario analysis are uncertain to some extent, because of some uncertain factors. The uncertainty analysis of scenarios is shown in Table 2. The percentages in Table 2 indicates the change rates of emissions from 2019 to 2050 due to the variation of corresponding factors.

**Table 2.** Uncertainty analysis of scenarios.

| Factor | Value | $CO_2$ | $SO_2$ | $NO_X$ | PM |
|---|---|---|---|---|---|
| Annual growth rate before 2030 | 4% | 2.28%~2.37% | 1.23%~1.33% | 0.45%~0.50% | 1.65%~2.01% |
| | 1% | −1.09%~−1.13% | −0.59%~−0.64% | −0.21%~−0.24% | −0.79%~−0.96% |
| Service life below 300 MW | 35 years | 0.04%~0.05% | 2.28%~2.34% | 1.16%~1.24% | 9.86%~10.11% |
| | 25 years | −0.04%~−0.05% | −2.01%~−2.08% | −1.12%~−1.20% | −9.74%~−10.01% |
| Substitution power | 75:25 (Large CPGs priority) | 0.03%~0.04% | 1.83%~1.96% | 1.66%~1.84% | 0.19%~0.23% |
| | 25:75 (Renewable energy priority) | −0.02%~−0.03% | −1.13%~−1.36% | −1.01%~−1.21% | −0.16%~−0.22% |

The first factor is the growth rate of power generation. If it is 4%, emissions will increase from 2019 to 2050. If it is 1%, emissions will decrease. The second factor is the service life of CPGs below 300 MW. Prolonging it to 35 years will increase each emission in different degrees. Shortening it to 25 years will reduce each emission in different degrees. The third factor is the structure of the substitution power, which represents the preference of power generation technology in future. More renewable energy to substitute the eliminated CPGs will decrease the emissions in future.

Considering lower economic growth rate and improvement of energy-saving technologies, the growth of power demand in the future will slow down, which will also effectively reduce emissions. The impact of the service life of small CPGs on emissions reduction should be paid attention to. Without considering the economic cost, it is better to eliminate backward CPGs as soon as possible. Besides, rapid development of renewable energy will bring about reductions of $SO_2$ and $NO_X$ emissions.

## 4. Conclusions and Discussion

The Chinese government has formulated a series of policies and plans to control and guide the development of the electric power sector, including those summed up as *Developing Large Units and Suppressing Small Ones*. The stock evolution analysis provides a new perspective to investigate the implementation effects of such policies and plans. Most previous studies took China's coal-fired power system as a whole, and hardly explored the structure effect of CPGs. Thus, they failed to connect the important policies with the development of CPGs in future. Some of them established dynamic

simulation models and proposed policy suggestions that do not tally with the facts, most likely because they ignored the update mechanism of CPG stock itself. After analysis of the stock evolution and environmental effects of China's CPGs in history and future, we obtained some useful findings.

The stock evolution of CPGs is featured that 600–1000 MW and 300–600 MW will dominate in coal-fired power systems and the CPGs of 100 MW and below will be eliminated in next few years.

The LMDI analysis indicates that the environmental benefits of structural evolution have been very limited up to now, compared with the changes in emission coefficient due to technical renovation. The sharp reductions of $SO_2$, $NO_X$, and PM emissions were due to strict enforcement of national emission standards and technology improvement.

CPGs will remain a major player in providing electricity for a long time. In the future, the installed capacities of CPGs in China will reach a peak and then start to decline, and the structure will be optimized. According to the TCT model and scenario analysis, the $CO_2$ emissions of CPGs will peak during 2025–2030 and decrease after that. Other pollutants will continue declining due to elimination of outdated CPGs. Uncertainty analysis indicates that reducing the growth rate of power generation, accelerating the elimination, and increasing the proportion of renewable energy in substitution power can effectively reduce the direct emissions of the coal-fired power sector. Therefore, promoting structural evolution by *Developing Large Units and Suppressing Small Ones* policies will be conducive to future emission reductions.

There are some limitations in this study. Only direct emissions of CPGs are considered, while the emissions of dismantling and construction are not. Some small-scale CPGs are used for heat and power cogeneration and new energy peak-shaving, and the proposal for accelerating elimination may not apply to these units. As the emission coefficients of different levels of CPGs in different years are estimated based on the monitoring data from State Grid Corporation, regional differences are not considered in TCT models and scenario analysis with such data. The economic cost in the structural evolution was not measured, thus the economic feasibility of accelerating elimination cannot be verified. In future, we will continue to make up for these shortcomings.

**Author Contributions:** F.L. conceived, designed the study and revised the final manuscript; Z.Y. analyzed the data and wrote the paper; X.X. analyzed the data and revised the manuscript; D.M. contributed primary data and analysis tools.

**Funding:** This research was funded by the National Natural Science Foundation of China (41401126) and State Grid Anhui Electric Power Corporation (SGAH0000FJJS1600572).

**Conflicts of Interest:** The authors declare no conflict of interest.

## Nomenclature

*Acronyms*

| | | | |
|---|---|---|---|
| CPGs | Coal-fired power generators | TCT | Type-cohort-time |
| LMDI | Logarithmic Mean Divisia Index | GB | National emissions standard |
| $CO_2$ | Carbon Dioxide | $SO_2$ | Sulfur Dioxide |
| $NO_x$ | Nitrogen Oxide | PM | Particulate Matter |

*Symbols*

| | | | |
|---|---|---|---|
| $C$ | the installed capacity of one CPG | $\tau$ | the average lifetime of CPGs |
| $t$ | the label of time | $\sigma$ | the standard deviation of CPGs' lifetime |
| $N$ | the number of CPGs | $\varepsilon$ | the emission coefficient |
| $S$ | the CPG stock | $T$ | the annual running hours |
| $\Delta S$ | the change in CPG stock | $E$ | the pollutant emission |
| $I$ | the new installed capacity | $\Delta E_S$ | the emission change due to total installed capacity change |
| $O$ | the installed capacity of eliminated CPGs | $\Delta E_R$ | the emission change due to installed structure change |
| $TS$ | the proportion in total stock | $\Delta E_T$ | the emission change due to annual running hours change |
| $O'$ | the adjusted installed capacity of eliminated CPGs | $\Delta E_\varepsilon$ | the emission change due to emission coefficient change |
| $f$ | the possibility of being eliminated | $w$ | the logarithmic mean weight |

## Appendix A

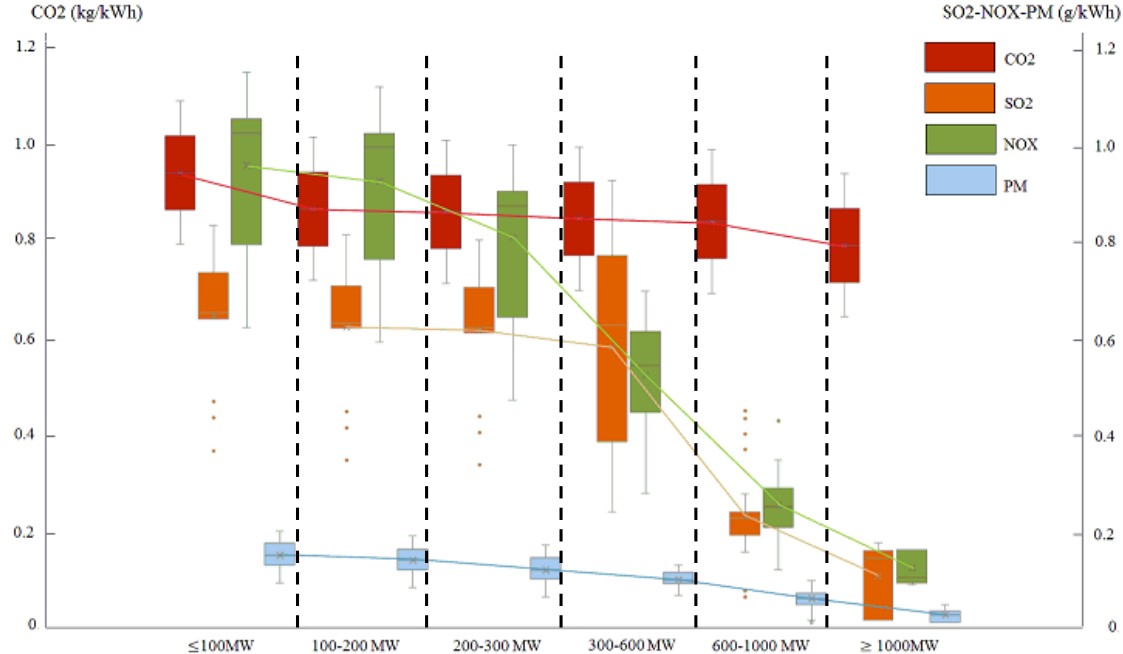

**Figure A1.** Emission coefficients distribution of CPGs in 2018.

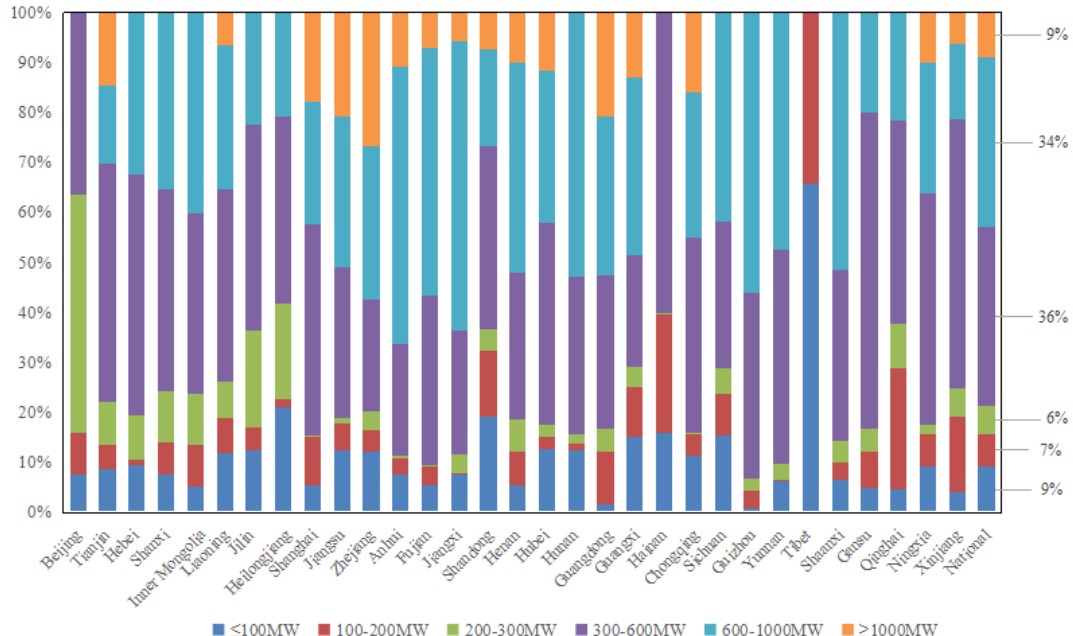

**Figure A2.** Stock structure of CPGs by province in 2018.

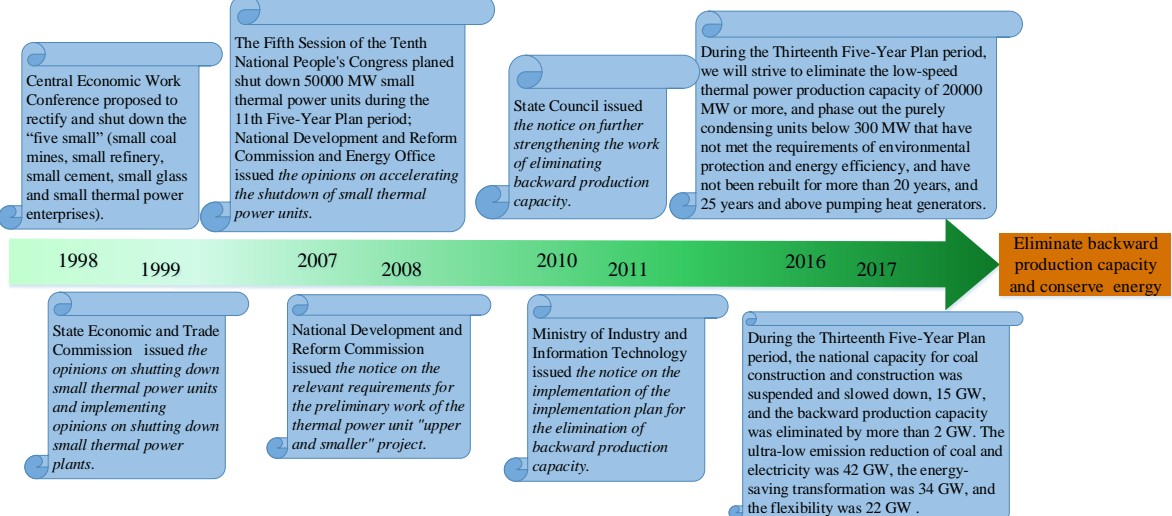

**Figure A3.** Policies related to CPG in China.

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
