# Peer review of "Environmental Benefits of Stock Evolution of Coal-Fired Power Generators in China"

_sustainability, doi:10.3390/su11195537_

Round 1

Reviewer 1 Report

The submission, entitled with "Environmental benefits of stock evolution of coal-fire power generators in China", estimated the costs and benefits. This review the following the comments.

At Line 118, should it be \delta S_{t}-O_{t}<0, not \delta S_{t}+O_{t}<0?

    2. (4) holding needs the condition of \delta S_{t}\leq O_{t}. This condition is not seen in the submission. 

     3. The 1st equality of (4) implies S_{\prime}_{t+1}=S_{t}-\delta S_{t+1}. Thus, \delta S_{t+1}=S_{t}-S_{\prime}_{t+1}. Together with (5) considering, we obtain:  S_{t}-S_{\prime}_{t+1}=S_{t+1}-S_{\prime}_{t}. Whether this is a fact is this reviewer's concern. This point will affect reading the following content.

      4. At Line 414, remove "and"; at the next line, "and" had better be changed into "but"; and at Line 417, "has" needs changing into "have".

       5. The final sentence should have a subject.

Author Response

Thank you for your valuable suggestions. Please see the attachment.

1.At Line 118, should it be \delta S_{t}-O_{t}<0, not \delta S_{t}+O_{t}<0?

Response: Thank you very much for this point. When delta S_{t}<0, it is possible that S_{t}+O_{t}<0.

2. (4) holding needs the condition of \delta S_{t}\leq O_{t}. This condition is not seen in the submission. 

Response: We have revised all the equations.

3. The 1st equality of (4) implies S_{\prime}_{t+1}=S_{t}-\delta S_{t+1}. Thus, \delta S_{t+1}=S_{t}-S_{\prime}_{t+1}. Together with (5) considering, we obtain:  S_{t}-S_{\prime}_{t+1}=S_{t+1}-S_{\prime}_{t}. Whether this is a fact is this reviewer's concern. This point will affect reading the following content.

Response: We have revised all the equations. The original process is from reference [14], and we made a modification.

[14]Vásquez, F.; Løvik, A.N.; Sandberg, N.H.; Müller, D.B. Dynamic type-cohort-time approach for the analysis of energy reductions strategies in the building stock. Energy & Buildings 2016, 111, 37-55.

4. At Line 414, remove "and"; at the next line, "and" had better be changed into "but"; and at Line 417, "has" needs changing into "have".

Response: We have revised these mistakes.

5.The final sentence should have a subject.

 Response: We have added a subject "we".

Reviewer 2 Report

The topic of the paper is interesting, but certain points require some illumination (as indicated below).

Authors are encouraged to introduce a nomenclature section at the beginning of the manuscript, including all variables and acronyms in the manuscript, in order to make the text more clear and readable. Authors can improve the abstract by including the existing challenges, motivations and outcomes of the paper. Moreover, the authors should illustrate and demonstrate the practical benefits of employing such an approach. The abstract requires major revisions to reflect the key ideas of this paper as well as the details of the proposed approach.

This paper is well structured, easy to read and the subject is important. But, in my opinion, more work must be done prior its publication. Introduction must be significantly extended and improved. The impact of global warming and CO2 emission reduction due to coal-fired power generation must be mentioned. Also, it is necessary to discuss the dynamic models. I suggest to study and introduce the following literature to the reference list (with certain topics): Economic, energy and environmental impact of coal-to-electricity policy in China: A dynamic recursive CGE study (Science of The Total Environment); Optimization Model of an Efficient Collaborative Power Dispatching System for Carbon Emissions Trading in China (Energies); Design of a Novel Voltage Controller for Conversion of Carbon Dioxide into Clean Fuels Using the Integration of a Vanadium Redox Battery with Solar Energy (energies); The dynamic linkage effect between energy and emissions allowances price for regional emissions trading scheme pilots in China (Renewable and Sustainable Energy Reviews); Dynamic operation and control of microgrid hybrid power systems (Energy); The role of multi-region integrated emissions trading scheme: a computable general equilibrium analysis (Applied Energy); Assessment of economic impacts of differentiated carbon reduction targets: a case study in Tianjin of China (Journal of Cleaner Production).

In section 2, the novelty of work is not clear. The type-cohort-time (TCT) and Logarithmic Mean Divisia Index (LMDI) methods are already available in the literature and in text books. Can you elaborate and explain more for your contributions? Maybe give some references. The involvement of the TCT and LMDI in the modelling deployed is mainly investigated on the environmental viewpoint of the research conducted. Which is the involvement upon the technical and the settings applied in the region of analysis? In Section 3, the results are lack of realistic study. The authors should consider severe operations.  Further investigation and comparisons can be conducted to identify the validation of the proposed method.  Based on 6, I suggest a comparison with the literature, in order to prove the efficiency of the proposed method? It can be seen from the result section that mostly results are not compared with the latest published papers. In Section 3, the authors are encouraged to provide a greater depth of discussion (more details), and modify the discussion and conclusion as well.

Author Response

Thank you for your valuable suggestions. Please see the attachment.

Round 2

Reviewer 1 Report

The authors have answered and responded to the comments. No other comments will be raised by this reviewer.

Author Response

Thank you!

Reviewer 2 Report

Good work, presenting the results and discussion, and analyzing the most important aspects. However, I still expect the authors to mention the technical literature for Conversion of Carbon Dioxide into Clean Fuels, and increase the connection with recommended literature in the first round.

Author Response

Thank you for your comments. We mentioned the technical literature for Conversion of Carbon Dioxide into Clean Fuels as literature [34], and increase the connection with recommended literature, such as literatures [25-33], [35-45].